# Large-scale epidemiology of opisthorchiasis in 21 provinces in Thailand based on diagnosis by fecal egg examination and urine antigen assay and analysis of risk factors for infection

Kulthida Y. Kopolrat[1,2], Chanika Worasith[1,3], Phattharaphon Wongphutorn[1,4], Anchalee Techasen[1,5], Chatanun Eamudomkarn[6], Jiraporn Sithithaworn[5], Watcharin Loilome[1,7], Nisana Namwat[1,7], Attapol Titapun[1,8], Chaiwat Tawarungruang[1,9], Bandit Thinkhamrop[1,9], Samarn Futrakul[10], Simon D. Taylor-Robinson[11], Melissa R. Haswell[12,13,14], Narong Khuntikeo[1,8†], Thomas Crellen[15,16,17]*, Paiboon Sithithaworn [1,6]*

1 Cholangiocarcinoma Research Institute, Khon Kaen University, Khon Kaen, Thailand, 2 Faculty of Public Health, Kasetsart University Chalermphrakiat Sakon Nakhon Province Campus, Sakon Nakhon, Thailand, 3 Department of Adult Nursing, Faculty of Nursing, Khon Kaen University, Khon Kaen, Thailand, 4 Biomedical Science Program, Graduate School, Khon Kaen University, Khon Kaen, Thailand, 5 Faculty of Associated Medical Sciences, Khon Kaen University, Khon Kaen, Thailand, 6 Department of Parasitology, Faculty of Medicine, Khon Kaen University, Thailand, 7 Department of Biochemistry, Faculty of Medicine, Khon Kaen University, Khon Kaen, Thailand, 8 Department of Surgery, Faculty of Medicine, Khon Kaen University, Khon Kaen, Thailand, 9 Data Management and Statistical Analysis Center (DAMASAC), Faculty of Public Health, Khon Kaen University, Khon Kaen, Thailand, 10 Department of Disease Control, Ministry of Public Health, The Office of Disease Prevention and Control 7 Khon Kaen, Khon Kaen, Thailand, 11 Department of Surgery and Cancer, St Mary's Campus, Imperial College London, Losndon, United Kingdom, 12 Indigenous Strategy and Services, University of Sydney, Sydney, New South Wales, Australia, 13 School of Geosciences, University of Sydney, Sydney, New South Wales, Australia, 14 School of Public Health and Social Work, Queensland University of Technology, Brisbane, Queensland, Australia, 15 Saw Swee Hock School of Public Health, National University of Singapore, Singapore, 16 School of Biodiversity, One Health and Veterinary Medicine, Graham Kerr Building, University of Glasgow, Glasgow, United Kingdom, 17 Nuffield Department of Medicine, Big Data Institute, University of Oxford, Oxford, United Kingdom

†Deceased
* paib_sit@kku.ac.th (PS); thomas.crellen@nus.edu.sg (TC)

## Abstract

### Introduction

Infection with the carcinogenic fish-borne trematode *Opisthorchis viverrini,* known as opisthorchiasis, is a major cause of biliary cancer (cholangiocarcinoma). Despite decades of disease prevention and control in Thailand, the parasite remains endemic. Here we apply a novel antigen assay for mass screening of opisthorchiasis and compare the prevalence against the conventional examination and analyze risk factors associated with current *O. viverrini* infection.

### Materials and methods

We conducted a large-scale cross-sectional survey to assess transmission of O. viverrini in the North, Northeast, and Eastern regions of Thailand. We screened

**Data availability statement:** All relevant data are within the manuscript and the data underlying the findings in this study are available in the S1 Data files (XLSX) in the supporting information.

**Funding:** This work described here was supported by the National Research Council of Thailand as a part of the Fluke-Free Thailand program (https://www.nrct.go.th). This study was partially funded by a Wellcome Trust Institutional Strategic Support Fund Grant (https://wellcome.org/grant) to Professor Narong Khunikeo, who sadly passed away in 2022. His immense contribution and leadership will never be forgotten. The support of the AMMF Cholangiocarcinoma Charity, led by Helen Morement (https://ammf.org.uk), and of the Kadoorie Foundation (Hong Kong, China), is also acknowledged. ST-R thanks the UK NIHR Biomedical Facility at Imperial College London for infrastructural support. This research was funded in whole, or in part, by the Wellcome Trust (Grant number 215919/Z/19/Z to TC). For the purpose of open access, the author has applied a CC BY public copyright license to any Author Accepted Manuscript version arising from this submission. The funders had no role in study design, data collection and analysis, decision to publish, or preparation of the manuscript.

**Competing interests:** The authors have declared that no competing interests exist.

randomly selected people (age 15 years and over) in 23 sub-districts, within 21 provinces, with a target sample size of 1,000 per sub-district. Each participant was screened for multiple helminth infection by fecal examination (quantitative formalin-ethyl acetate concentration technique; FECT), and the antigen assay by monoclonal antibody-based enzyme-linked immunosorbent assay (ELISA) was applied to urine samples to detect O. viverrini. We collected risk factors for O. viverrini infection using standardized questionnaire surveys. The data were analyzed with regression models which correlated individual-level explanatory variables against i) infection status with O. viverrini and ii) the intensity of infection, as measured by the antigen assay or FECT.

## Findings

Of the 20,322 individuals enrolled, 19,465 provided urine samples for antigen detection by ELISA and 18,929 provided fecal samples for examination by FECT. The urine antigen assay revealed an overall opisthorchiasis prevalence of 50.3%, a fourfold increase over the 12.2% prevalence detected by FECT. Marked spatial heterogeneity was observed, with antigen-based prevalence estimates ranging from 22.2% to 71.4% and several localities exceeding 60%. When assessed against a composite reference standard (combined ELISA and FECT), the urine ELISA yielded a diagnostic sensitivity of 91.6%, compared with 21.9% for FECT. We found a positive correlation between fecal egg counts and the concentration of worm antigen in urine across study sites. The ratio between the prevalence of O. viverrini observed by the antigen assay and FECT was high in provinces with a low mean number of O. viverrini eggs, and the ratio approached unity as the mean eggs per gram of stool (EPG) increased. Similar aggregate distribution patterns of fecal egg counts (EPG) and urine antigen concentrations suggest that the urine assay has potential for quantitative diagnostic evaluations. When analyzing individual-level risk factors, we further identified age, sex, occupation, a history of prior treatment with praziquantel, history of O. viverrini examination, and raw fish consumption as predictive of infection with O. viverrini, while a higher education level and certain occupations emerged as protective factors.

## Conclusions and recommendations

Application of the antigen assay to diagnose O. viverrini infection yielded a four-fold higher prevalence than the fecal egg examination, with the highest difference in low endemicity regions, which suggests that previous surveys may have underestimated the extent of opisthorchiasis in Thailand. Given the ease of urine sample collection, our study highlights the potential for application of the antigen assay as a new tool in the control of opisthorchiasis.

Diseases

## Author summary

Opisthorchiasis is a neglected tropical disease caused by an infection with the foodborne liver fluke, *Opisthorchis viverrini*. The parasite is a group one carcinogen, and chronic infection leads to an aggressive cancer of the liver, cholangiocarcinoma, which is generally fatal. A prerequisite for the prevention of cholangiocarcinoma in *O. viverrini* endemic regions is, therefore, accurate screening and treatment to prevent and treat liver fluke infections and reduce community-wide transmission. Mass screening for opisthorchiasis typically relies on microscopic detection of parasite eggs in stool, which is known to have a low sensitivity. Hence the observed prevalence is likely to be underestimated in parasitological surveys. Here, we applied a novel and sensitive worm antigen assay to urine samples and compared the results against the standard fecal egg examination to assess the endemicity of opisthorchiasis in Thailand. Our survey results showed a markedly higher prevalence of *O. viverrini* by urine assay (50.3%) compared with fecal examination (12.2%). Both diagnostic methods yielded similar prevalence profiles when stratified by the participant's age and sex, with a higher risk of infection among men and in older age groups. We found agreement between the two diagnostic methods for the quantitative intensity of infection, and the urine assay detected more cases than fecal examination when the intensity of parasite infection is low, which constitutes the majority of currently infected people in Thailand. As providing urine samples is generally more acceptable among the public than stool sample collection, the antigen assay has the potential to be a valuable diagnostic tool for mass screening of opisthorchiasis, leading to improved disease prevention and control.

## Introduction

Despite decades of disease control programs, infection with the carcinogenic trematode *Opisthorchis viverrini* remains widespread across Southeast Asia including Thailand, Lao PDR, Cambodia, Vietnam, and Myanmar with estimated >12 million human cases [1–3]. Human infection with *O. viverrini* arises from the consumption of raw, or insufficiently cooked, freshwater fish containing infective metacercariae and the cycle of transmission is perpetuated by parasite eggs in human or animal feces contaminating freshwater sources containing *Bithynia* snails [4]. Long-term infection with the parasite can lead to chronic inflammation, hepatobiliary morbidity, and eventually cholangiocarcinoma [5,6]. Since 1994, infection with *O. viverrini*, and the related liver fluke, *Clonorchis sinensis,* have been designated as group one carcinogens [7], causing bile duct cancer (cholangiocarcinoma; CCA) in humans. Hence, the primary prevention strategy for CCA in liver fluke endemic settings is to treat people infected with the *O. viverrini* parasite and halt community-wide transmission [8–11].

In Thailand, the spatial distribution of opisthorchiasis based on fecal egg detection is well defined, with the highest prevalence and intensity of infection found in the Northeast region and with limited foci of infection also found in the Central and Northern regions [1,6]. The first national screening for liver fluke in Thailand between 1980–81 reported an overall prevalence of 14.7%, with the Northeast, North, Central, and South regions reporting prevalences of 34.6%, 5.6%, 6.3%, and 0.11%, respectively [1]. A national liver fluke control program focused on egg detection and the treatment of infected individuals with the highly effective drug, praziquantel from 1987-1991. Subsequent nationwide surveys detected declines in the prevalence at a national level, to 11.8% in 1996 and 9.6% in 2001, and in Northeast Thailand, to 12.4% in 1996 and 15.7% in 2001 [12,13]. Subsequently, in 2014 the national survey showed that sustained declines in prevalence had taken place across Thailand, with an overall prevalence of 5.1% nationally and 9.2% in the Northeast region [2].

Notably, these surveys for *O. viverrini* relied on the conventional method of parasite egg detection using stool microscopy techniques, such as the formalin-ether concentration technique (FECT) or the Kato-Katz thick smear. In the current transmission landscape of *O. viverrini*, these fecal examination diagnostics may have low accuracy because of sparse egg output from low worm burdens [8], potential misdiagnosis due to morphologically similar eggs from minute intestinal

flukes, and the need for experienced microscopists [14,15]. Molecular methods to discriminate these parasites, such as PCR, have been developed [16], however high costs prevent application in mass screening in resource-limited settings.

An alternative approach to increase diagnostic sensitivity for opisthorchiasis is to detect antigens secreted by adult worms. Initial studies have highlighted the potential of copro-antigen detection [17,18], although this technique requires the collection of stool samples, which may be undesirable for the study population and risks exposure of technicians to fecal pathogens. More recently, a new approach for antigen detection using urine samples was found to be more sensitive than fecal egg examination for screening and elimination of opisthorchiasis following treatment with the anthelmintic, praziquantel [19,20]. The persistent levels of antigen in urine over time in the absence of treatment and low daily variability provide further evidence to support the reliability of the urine assay for diagnosis and population screening [21,22]. Based on the available evidence that antigen detection is highly sensitive, especially in areas with low intensity of infection, however, its use in a larger-scale study to assess the current regional status of opisthorchiasis has not been reported. We hypothesize that the estimated prevalence of opisthorchiasis will be 2–4 times higher in large surveys by the urine antigen test compared against fecal egg examination.

This study was conducted as a part of the national program to control liver fluke and cholangiocarcinoma (CCA) in Thailand launched in 2016. The aim was to create the cohort population in the endemic communities in Thailand for a longitudinal study with testing and treatment performed at multiple time points to control liver fluke and CCA in Thailand. The specific aims of this study were to a) establish the baseline prevalence and infection intensity of *O. viverrini,* as measured by fecal egg examination and the urine antigen assay; b) analyze the relationship between prevalence and intensity of *O. viverrini*; and c) determine demographic and behavioral risk factors associated with *O. viverrini* infection, as diagnosed by both urine antigen test and fecal egg examination.

## Materials and methods

### Ethics statement

The human subject protocol used in this study was approved by the Ethics Committee of Khon Kaen University, Khon Kaen, Thailand (reference HE601370). The project coordination and field operation were performed in collaboration with the Ministry of Public Health (MOPH), provincial, district, and sub-district health offices. Prior to enrollment, all participants received an explanation of the study goals and study procedures. All participants aged 18 years and over provided written informed consent, while parents or legal guardians provided verbal consent for participants aged 15–17 years. Treatment with a single oral dose of praziquantel (40 mg/kg of body weight) was given to participants who were diagnosed as *O. viverrini* positive by FECT or urine antigen detection assay. Participants who were found positive for other parasitic infections were given appropriate anthelmintic drugs.

Experimental protocols involving laboratory animals (mice and rabbits) were approved by the Institutional Animal Ethical Committee, Khon Kaen University (AEKKU 24/2558), and were specifically used for antibody production in this study. On necropsy day, animals were euthanized with isoflurane prior to cardiac blood collection to obtain antibodies. Humane endpoints were implemented with daily health assessments of the animals to ensure their well-being. The procedure adhered strictly to the guidelines set forth by the National Research Council of Thailand for the Care and Use of Laboratory Animals.

### Project design and sample population

A cross-sectional, community-based study was conducted during November 2017-September 2018 in an endemic area for infection with *O. viverrini*. This survey also represents the baseline of a multi-year cohort study to investigate liver fluke transmission and control of CCA in Thailand. The participants were the local population in 21 provinces of North, East, and Northeast Thailand. A household random sample of the sub-district population aged 15 years old was contacted

and invited to participate in the project (S1 Text, Sample size calculation). Younger individuals aged <15 years were not recruited as a strategy to focus on groups with a higher likelihood of infection for longitudinal study. In this study, the total number of enrolled participants were 20,322 from two provinces in the Northern region (Chiang Mai [CMI] and Phayao [PYO]), one province in the Eastern region (Sa Kaeo [SKW]), and eighteen provinces in the Northeast region (Khon Kaen [KKN], Roi Et [RET], Kalasin [KSN], Maha Sarakham [MKM], Udon Thani [UDN], Nong Bua Lamphu [NBP], Chaiyaphum [CPM], Loei [LEI], Sakon Nakhon [SNK], Nakhon Phanom [NPM], Bueng Kan [BKN], Nong Khai [NKI], Nakhon Ratchasima [NMA], Ubon Ratchathani [UBN], Surin [SRN], Si Sa Ket [SSK], Buriram [BRM], and Yasothon [YST]) (Fig 1).

## Demographic data collection

In addition to screening parasitic infections, including *O. viverrini*, the participants were asked to answer questions in a questionnaire containing socio-demographic information (age, gender, level of education, main occupation), history of *O. viverrini* infection, history of previous praziquantel treatment, as well as other health and lifestyle information.

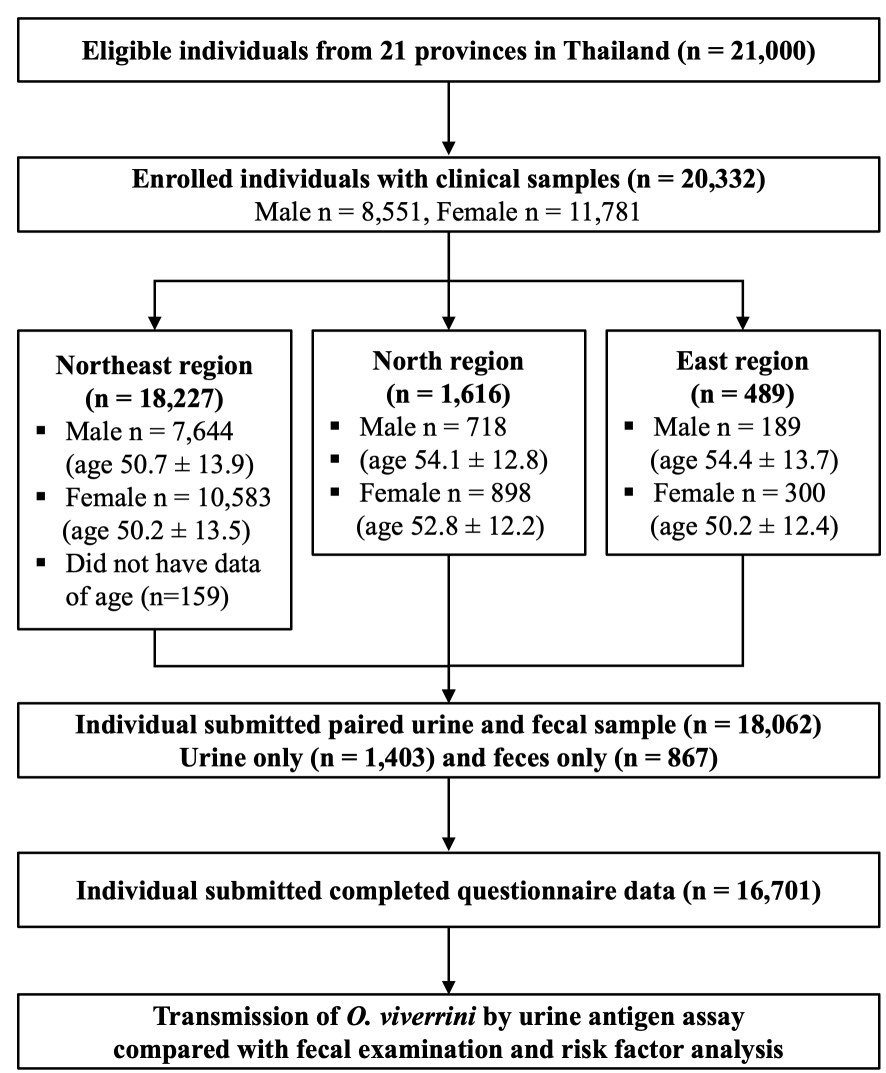

**Fig 1. Flow chart of study participants aged 15 years and over from 21 provinces in Thailand.**

## Clinical sample collection

Clean plastic containers labeled with identifying numbers were distributed to the project participants to collect samples, which included feces (approx. 10 grams) and the first morning, mid-stream urine samples (approx. 10 ml). Fecal samples were kept in containers at ambient temperature, while the urine samples were kept in a chilled insulated box and transported from the study site to the laboratory at Khon Kaen University within one day of collection. At the laboratory, fecal samples were weighed and fixed in 10% formalin and processed for parasite examination using the formalin-ethyl acetate concentration technique (FECT). Urine samples were centrifuged at 988 × $g$ at 4 °C for 15 minutes and the supernatants were aliquoted and stored at -20 °C until used for the urine assay.

## Fecal examination by quantitative formalin-ethyl acetate concentration technique (FECT)

For parasitological diagnosis, the quantitative formalin-ethyl acetate concentration technique (FECT) was performed as previously described [23]. Two grams of fresh stool were processed by fixing with 10% formalin and thoroughly shaken before being strained through gauze. Three milliliters of ethyl acetate were added to the mixture to extract fat from the feces. After vigorous shaking and centrifugation at 1,455 × $g$ for 5 minutes, the supernatant was discarded, and the remaining matter was re-suspended in 10% formalin. A drop of fecal suspension was placed on a slide and examined, and enumeration of parasites was done in duplicate. The intensity of infection was calculated by the number of eggs counted per drop examined divided by 2 (grams of stool) and multiplied by the total drops (volume) of fecal suspension. Discrimination between eggs of *O. viverrini* and minute intestinal fluke (MIF), for instance, *Phaneropsolus bonnei* or *Prostodendrium molenkapi*, was performed based on morphological characteristics as previously described [15,24].

## Urine assay by monoclonal antibody-based enzyme-linked immunosorbent assay (ELISA)

The protocol for the *O. viverrini* antigen assay in urine using monoclonal antibody-based ELISA was described previously [19]. Briefly, polystyrene microtitration plates (Nunc, Roskilde, Denmark) were coated with 5 µg/mL of the monoclonal antibody (specific to the antigen of *O. viverrini*, clone KKU 505) diluted in 50 mM bicarbonate buffer (pH 9.6). The plates were sealed and incubated at night at 4°C. The next day, plates were washed three times with a buffer containing 0.05% Tween 20 in PBS pH 7.4 (PBST) and uncoated sites were blocked with 5% dried skimmed milk in PBST. The plates were then incubated at 37 °C for 1 hour. Washing was repeated thrice with PBST, and samples of undiluted urine pre-treated with TCA (100 µl/well) were added to wells in duplicate and incubated at 37 °C for 2 hours. The plates were washed 5 times with PBST, and IgG rabbit anti-crude *O. viverrini* antigen was added and incubated at 37 °C for 1 hour. After three washes, 1:4,000 diluted biotinylated goat anti-rabbit IgG (Invitrogen, Carlsbad, CA, USA) in PBST was added and incubated at 37°C for 1 hour. Thereafter, the plates were washed three times, and streptavidin horseradish peroxidase (HRP)-conjugate (GE Healthcare, Buckinghamshire, United Kingdom) (1:5000 dilution, 100 µl/well) was added. After incubation and washing, a substrate solution (o-phenylenediamine hydrochloride) solution (Sigma, St. Louis, MO, USA) (100 µL/well) was added, and plates were incubated for 20 min in the dark at room temperature. The reaction was stopped by the addition of 2M sulfuric acid ($H_2SO_4$), and the plates were read on an absorbance reader (Tecan, Grödig, Austria) at the optical density (OD) of 492 nm.

The OD values of urine analysis from proven cases of positive and negative *O. viverrini* infection ($n = 40$/group) determined by FECT were used to construct a receiver operation curve (ROC). The cut-off point of OD > 0.237 was considered as a positive diagnosis by urine ELISA using MedCalc software version 9.6.3 (MedCalc, Ostend, Belgium). In order to quantify the antigen concentration in the urine sample, the standard curve was constructed based on spiked urine with the crude antigen of *O. viverrini* and measured by ELISA. The relationships between the urinary antigen concentrations (X) and OD values (Y) from ELISA were estimated by the best fit linear regression equation (Log Y = 0.941X − 0.914). The OD

values were transformed into concentrations of *O. viverrini* antigen in urine using standard curves and expressed as ng/mL. Based on the cut-off OD value, a sample was considered positive when antigen concentration in urine was > 16.7 ng/mL.

**Data management and statistical analysis**

Data obtained in this project were collected and managed under the ethical procedure of the project by the Isan cohort database, as previously described [25]. Demographic information and responses to questions about potential risk factors from the questionnaires and laboratory data were entered into an Excel worksheet (Microsoft) and analyzed using SPSS 26 (IBM, Chicago, IL, USA) and R version 4.0.2 (R core team Vienna, Austria, 2018). Baseline characteristics of the sample were presented as frequency numbers and percentages for categorical data. The continuous data were described using mean, median, and standard deviation (SD). The prevalence of *O. viverrini* infections was estimated overall and separately for each category of factors, including gender and age groups. Seven age groups were established as follows: (i) <20 years, (ii) 20–29 years, (iii) 30–39 years, (iv) 40–49 years, (v) 50–59 years, (vi) 60–69 years, and (vii) ≥ 70 years. Thematic maps were used to depict the spatial distribution of *O. viverrini* infection using Quantum GIS (https://www.qgis.org/en/site/). The prevalence map was created in R software using shape files from UN-OCHA (https://data.humdata.org/dataset/geoboundaries-admin-boundaries-for-thailand) which are licensed under a Creative Commons Attribution 4.0 International license.

Diagnostic accuracy of urine antigen detection by ELISA and FECT for opisthorchiasis in terms of sensitivity, specificity, positive predictive values (PPV), negative predictive values (NPV), and 95% confidence intervals (95% CI) for *O. viverrini* infection were determined using MedCalc (Med Calc Software, Ostend, Belgium). The agreement in the status of *O. viverrini* infections was evaluated using Cohen's kappa coefficient. Cohen's kappa values < 0 were interpreted as indicating no agreement between methods: 0–0.20 as slight, 0.21–0.40 as fair, 0.41–0.60 as moderate, 0.61–0.80 as substantial and 0.81–1 as almost perfect agreement on the status of *O. viverrini* infection between different methods [26]. Univariable logistic regression was run to determine the association between *O. viverrini* infection and questionnaire responses, including gender, age, educational levels, occupational status, history of *O. viverrini* examination, history of *O. viverrini* infection, history of praziquantel treatment, family history with liver cancer, smoking history, alcohol consumption history, alcoholism, and history of raw fish eating. Subsequently, factors for which the univariable logistic regression models had a p-value <0.25 were included in a multivariable logistic regression model. A backward variable selection method was used to determine the most parsimonious model. Odds ratio (OR) with 95% confidence interval (CI) was used to assess the strength of association between variables. Statistically, significance was considered at a 95% confidence level and p-value less than 0.05. The corrections to the Type I error for multiple comparisons were implemented to ensure confidence in the reported results.

## Results

### Prevalence of *Opisthorchis viverrini* by fecal examination and urine assay.

Based on examination by FECT, the overall prevalence of *O. viverrini* was 12.2% (n = 18,929), with variation between regions from 3.6% in the northern region to 13.1% in the northeast region (Table 1). The urine antigen test diagnosed many more cases of infection with *O. viverrini* (50.3%; n = 19,465) compared with the FECT diagnostic. The lowest prevalence was 49.3% in the northern region, and the highest prevalence was 59.3% in the eastern region (Table 1).

The overall prevalence of parasitic infection based on fecal examination by FECT was shown in Table 2 (n = 18,929). The most common species was *O. viverrini* and followed by infection with the roundworm *Strongyloides stercoralis* which had a prevalence of 4.4%. Other infections with minute intestinal flukes, tapeworms, and soil-transmitted helminths were found in the cohort at a combined prevalence of <2.0%.

**Table 1. Characteristics of study participants and *O. viverrini* infection by fecal examination or urine assay.**

| Region in Thailand | Northeast | North | East | Total |
|---|---|---|---|---|
| **No. of provinces** | 18 | 2 | 1 | 21 |
| **No. of participants** | 18,227 | 1,616 | 489 | 20,332 |
| **Gender** (mean of age ± SD) | | | | |
| Male | 7,644 | 718 | 189 | 8,551 |
| | (50.7 ± 13.9) | (54.1 ± 12.8) | (54.4 ± 13.7) | (51.1 ± 13.9) |
| Female | 10,583 | 898 | 300 | 11,781 |
| | (50.2 ± 13.5) | (52.8 ± 12.2) | (50.2 ± 12.4) | (50.4 ± 13.3) |
| **Age (years)** | | | | |
| ≤ 20 | 833 | 37 | 14 | 884 |
| 21-30 | 668 | 50 | 20 | 738 |
| 31-40 | 2,046 | 110 | 31 | 2,187 |
| 41-50 | 5,053 | 386 | 156 | 5,594 |
| 51-60 | 5,136 | 618 | 152 | 5,906 |
| 61-70 | 3,493 | 312 | 82 | 3,887 |
| > 70 | 839 | 103 | 34 | 976 |
| **FECT** | | | | |
| Tested (n) | 17,137 | 1,523 | 269 | 18,929 |
| Prevalence (%) | 2,243 (13.1) | 55 (3.6) | 11 (4.1) | 2,309 (12.2) |
| Intensity (log geometric mean ± SE) | 1.24 ± 0.01 | 0.60 ± 0.05 | 1.18 ± 0.14 | 1.23 ± 0.01 |
| **Urine ELISA** | | | | |
| Tested (n) | 17,482 | 1,602 | 381 | 19,465 |
| Prevalence (%) | 8,780 (50.2) | 789 (49.3) | 226 (59.3) | 9,795 (50.3) |
| Intensity (log geometric mean ± SE) | 1.68 ± 0.004 | 1.55 ± 0.01 | 1.82 ± 0.03 | 1.68 ± 0.004 |

**Table 2. Prevalence of *Opisthorchis viverrini* infection and other intestinal parasites in the sampling of sub-district communities across 21 provinces of Thailand (*n* = 18,929).**

| Parasites | No. of positive | Prevalence (%) |
|---|---|---|
| *Opisthorchis viverrini* | 2,309 | 12.2 |
| *Strongyloides stercoralis* | 830 | 4.4 |
| Minute intestinal flukes | 346 | 1.8 |
| *Taenia* sp. | 147 | 0.8 |
| Echinostomes | 113 | 0.6 |
| Hookworms | 36 | 0.2 |
| *Trichuris trichiura* | 9 | 0.05 |

The prevalence and distribution of *O. viverrini* by province determined by FECT and the urine antigen assay at the provincial level in Thailand using matched cases with fecal and urine samples were shown in Fig 2 (n = 18,062). Based on FECT, the prevalence varied with province from the lowest of 1.5% in Chiang Mai (CMI) and the highest of 30.2% in Kalasin (KSN). There were 4 provinces (19.0% of the study population) with a prevalence >20%, 6 provinces (28.6% of the study population) had a prevalence between 10% – 20%, and the rest (52.4% of the study population) had < 10% prevalence. Based on the urine antigen assay, the highest prevalence was 71.4% in Sa Kaeo (SKW) and the lowest was 22.3% in Khon Kaen (KKN). All provinces had a prevalence of *O. viverrini* with the antigen test >20.0%. In Northeast Thailand,

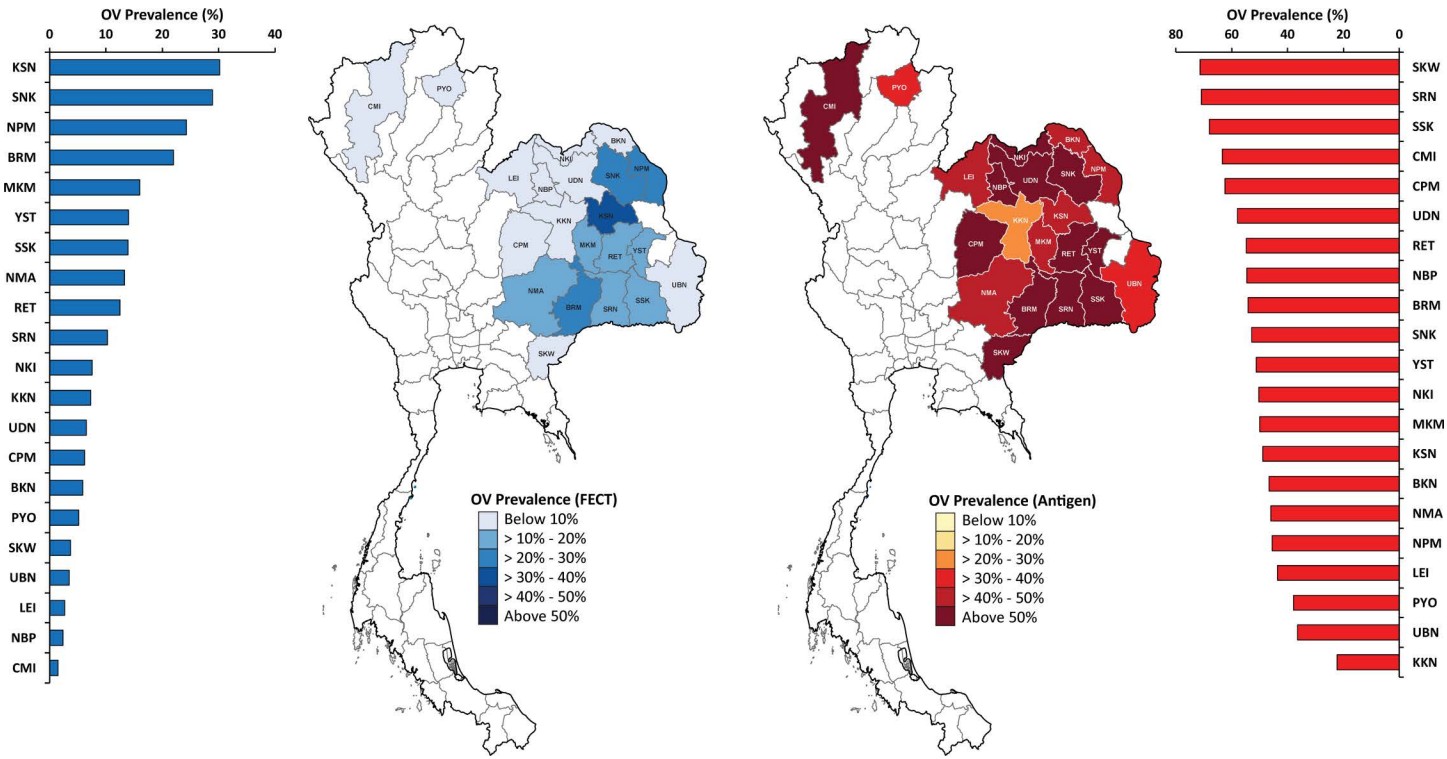

**Fig 2. Distribution of *O. viverrini* prevalence at provincial level in Thailand measured by FECT (A) and urine antigen assay (B).** The map showed the geographical location of the study sites in 21 provinces in Thailand. The horizontal bar graphs showed the prevalence of each study site. Northern region; CMI: Chiang Mai and PYO: Phayao, Eastern region; SKN: Sa Kaeo. Northeast region; KKN: Khon Kaen, RET: Roi Et, KSN: Kalasin, MKM: Maha Sarakham, UDN: Udon Thani, NBP: Nong Bua Lamphu, CPM: Chaiyaphum, LEI: Loei, SNK: Sakon Nakhon, NPM: Nakhon Phanom, BKN: Bueng Kan, NKI: Nong Khai, NMR: Nakhon Ratchasima, UBN: Ubon Ratchathani, SRN: Surin, SSK: Si Sa Ket, BRM: Buriram, and YST: Yasothon. The prevalence map was created in R software using shape files from UN-OCHA https://data.humdata.org/dataset/geoboundaries-admin-boundaries-for-thailand which are licensed under a Creative Commons Attribution 4.0 International license.

the mean prevalence in 18 localities was 13.1% by FECT and 50.2% by urine assay. The comparative prevalence by urine antigen assay and FECT at each study province was shown in S1 Fig.

**Age and gender-prevalence and intensity of *O. viverrini*.**

The prevalence of *O. viverrini* significantly increased with the age of the participants in both men and women, as diagnosed by FECT (Chi-square test for trend = 22.71-27.93; $p < 0.001$) (Fig 3). For the urine antigen assay, the relationship between age and prevalence was non-linear, as the prevalence for both men and women peaked at 40–49 years and then dropped for older age groups, giving a non-significant test for trend (Chi-square test for trend = 0.07-0.92; $p > 0.05$). The prevalence in men was higher than in women, as diagnosed by FECT and the urine assay (Chi-square test = 68.24-142.92, $p < 0.001$). The prevalence measured by urinary antigen detections was 4–5 folds higher than FECT when aggregated by age and sex. The prevalence map was created in R software using shape files from UN-OCHA https://data.humdata.org/dataset/geoboundaries-admin-boundaries-for-thailand which are licensed under a Creative Commons Attribution 4.0 International license.

The intensity of *O. viverrini* infection (egg/g feces) by FECT was significantly associated with age in both males and females (Kruskal-Wallis test; $p < 0.001$) (Fig 4A).

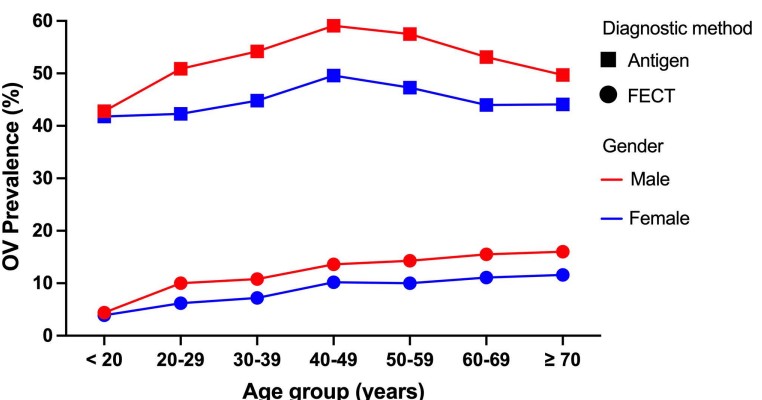

**Fig 3. Age-sex-prevalence profiles of *O. viverrini* infection determined by FECT and urine antigen assay.**

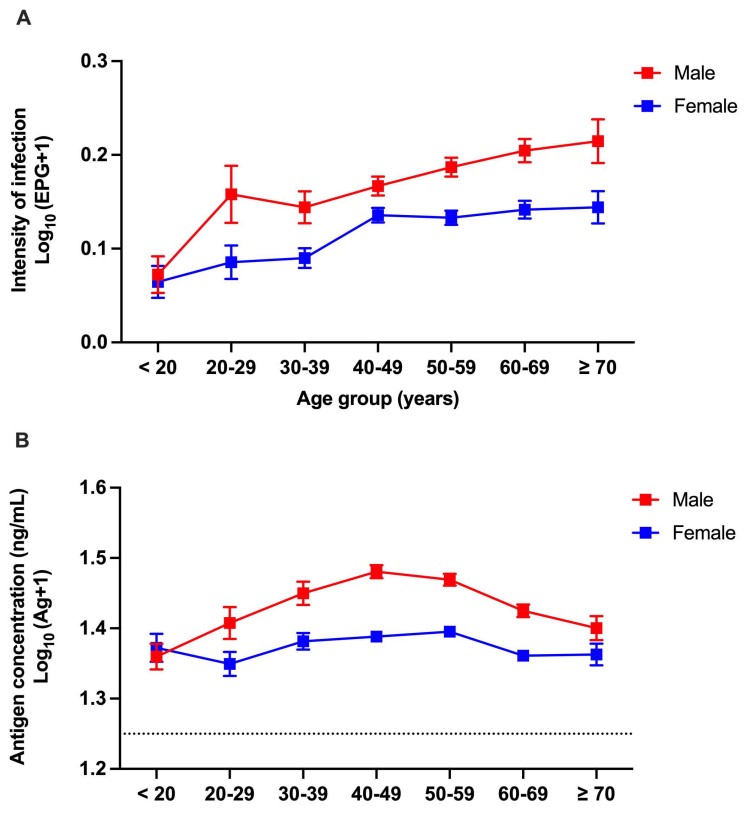

**Fig 4. Age-sex-intensity profiles of *O. viverrini* (fecal egg count) determined by FECT (A) and urine antigen assay (antigen concentration in urine) (B).**

In case of urine assay, the association with age was observed in both males and females by urine antigen detection (Kruskal-Wallis test; p < 0.001, p < 0.01, respectively) (Fig 4B).

**Relationships between fecal examination and urine assay.**

For quantitative diagnosis, the concentrations of urinary antigen showed a significant positive correlation with increasing intensity of *O. viverrini* (EPG) (Kruskal-Wallis test, p < 0.0001; Fig 5). The relationship between the prevalence ratios of antigen detection versus FECT against the intensity of *O. viverrini* infection was shown in Fig 6 when aggregated by study province. The ratio increased steeply at low intensity of infection, i.e., 1–5 EPG, and reached unity at 10–20 EPG onwards.

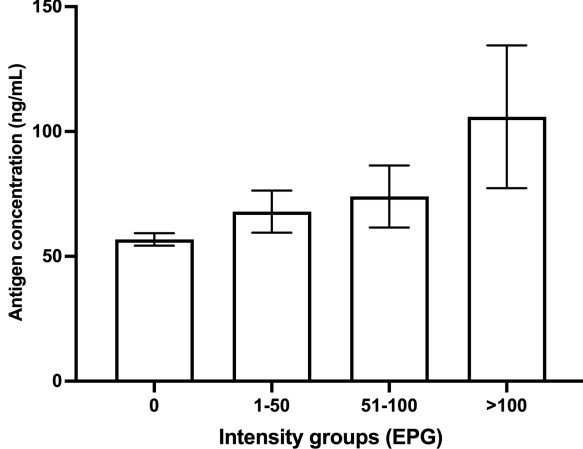

**Fig 5. Relationship between concentrations of antigen in urine (ng/mL) and intensity of *O. viverrini* (EPG) of individuals from Northeast Thailand.** Data shown were mean and standard variation (SD) of antigen concentrations in urine (ng/mL) for the Y-axis and the intensity group of *O. viverrini* based on fecal egg counts (egg/g feces) for the x-axis (Kruskal-Wallis test, p < 0.001).

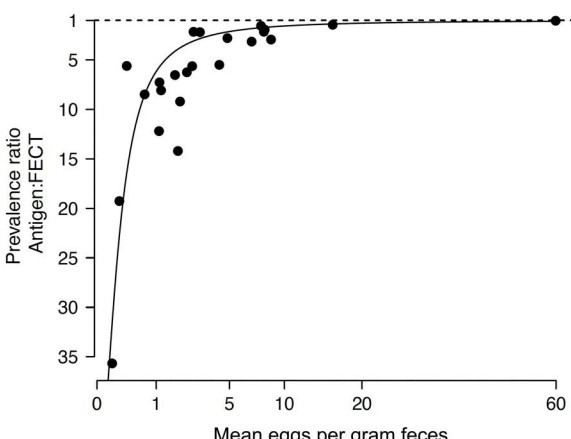

**Fig 6. The prevalence ratio between urine antigen assay and fecal egg counts.** The ratio varied from 1-35 folds, and the ratio trend approached unity at mean egg per gram feces >10-20 (Log-transformed values).

### Distribution of fecal egg count and urine antigen concentration.

The frequency distribution of fecal egg counts in the study participants exhibited aggregated or skewed distribution in which most participants had zero and low egg counts (1–20 EPG), and few people had a high-intensity infection (EPG > 500) (S2A Fig). A similar skewed distribution pattern was observed for antigen concentration in urine. The majority of participants were in the negative range (<16.7 ng/mL) and had low antigen concentrations (<100 ng/mL) (S2B Fig). The geometric mean antigen in the positive test was 47.8 ng/mL urine.

### The diagnostic accuracy

Diagnostic performance was evaluated using matched urine and fecal samples from 18,062 individuals (Table 3). Among participants who tested positive by urine ELISA, 49.2% of these were negative by FECT. Conversely, 38.2% of individuals with FECT-positive results tested negative by urine ELISA. The proportion of participants who were concurrently positive by both FECT and urine ELISA was 7.5%. Using FECT as the primary reference standard, the urine ELISA had 61.7% sensitivity, 50.6% specificity, 14.6% positive predictive value and 90.5% negative predictive value. When the combined FECT and urine ELISA was used as a composite reference standard, the sensitivity and specificity for urine ELISA were 91.6% and 100%, respectively, while FECT showed sensitivity of 21.9% and specificity of 22.7% and 100%, respectively. The observed agreement between the composite reference standard and urine ELISA was almost perfect (kappa = 0.9), whereas the agreement for FECT was slight (kappa = 0.2).

**Table 3. Diagnostic accuracy of urinary antigen detection by enzyme-linked immunosorbent assay (ELISA) and formalin ethyl-acetate concentration technique (FECT) against a primary reference standard (FECT) (a) and composite reference standard (combined FECT and ELISA) (b).**

**a) Primary reference standard (FECT)**

| Method | Primary std | | Sensitivity (95%CI) | Specificity (95%CI) | PPV) (95%CI) | NPV (95%CI) |
|---|---|---|---|---|---|---|
| | Pos (n) | Neg (n) | | | | |
| ELISA | | | | | | |
| Pos (n) | 1359 | 7830 | 61.7 (59.6-63.7) | 50.6 (49.8-51.4) | 14.6 (14.2-15.1) | 90.5 (90.0-91.0) |
| Neg (n) | 843 | 8030 | | | | |
| Total | 2202 | 15860 | | | | |

**b) Composite reference standard (combined FECT and ELISA)**

| Method | Composite std | | Sensitivity (95%CI) | Specificity (95%CI) | PPV (95%CI) | NPV (95%CI) |
|---|---|---|---|---|---|---|
| | Pos (n) | Neg (n) | | | | |
| ELISA | | | | | | |
| Pos (n) | 9189 | 0 | 91.6 (91.0-92.1) | 100 (99.9-100) | 100 (99.9-100) | 90.5 (89.9-91.0) |
| Neg (n) | 843 | 8030 | | | | |
| Total | 10032 | 8030 | | | | |
| FECT | | | | | | |
| Pos (n) | 2202 | 0 | 21.9 (21.1-22.8) | 100 (99.9-100) | 100 (99.8-100) | 90.3 (90.3-90.3) |
| Neg (n) | 7830 | 8030 | | | | |
| Total | 10032 | 8030 | | | | |

## Analysis of risk factors of *O. viverrini* infection

In the multivariable logistic regression model, where the outcome variable was a positive diagnosis by either FECT or urine ELISA; i) sex, ii) age, iii) occupational status, iv) history of *O. viverrini* examination, and v) history of raw fish eating, were all significantly associated with *O. viverrini* infection (Table 4). The odds of infection with *O. viverrini* were 1.49 (95% CI: 1.39-1.59, p < 0.001) times higher in males than in females. Older age groups had higher odds of infection, with an elevated odds ratio of 1.26-1.31 for infection, compared against participants less than 20 years of age. For occupation, farmers had higher adjusted odds of *O. viverrini* infection than other occupations (non-farmers) (aOR 1.14, 95% CI: 1.04-1.24, p = 0.004).

Compared with participants who had a history of *O. viverrini* examination, the aOR for *O. viverrini* infection among individuals without a history of examination was 1.12 (95% CI: 1.06-1.19, p < 0.001), and higher for those who had discovered *O. viverrini* eggs (aOR=1.24, 95% CI: 1.14-1.34, p < 0.001). People who had received previous treatment for opisthorchiasis had a higher risk of current *O. viverrini* infection compared with people who had not previously received anthelmintic treatment (aOR=1.21, 95% CI: 1.06-1.39, p = 0.006). Moreover, eating raw fish was associated with an increased risk of infection. Those who reported having eaten raw fish in the past or currently had an odds ratio of 1.11 (95% CI: 1.02-1.20, p = 0.011) for *O. viverrini* infection compared to those who reported never having eaten raw fish. A negative association was found between *O. viverrini* infection and higher educational levels, with people who had higher educational backgrounds (secondary school or certificate and higher) had significantly lower odds of infection than those with primary school or lower education (aOR = 0.80, 95% CI: 0.67-0.95, p = 0.013). A family history of liver cancer, smoking, and alcohol consumption were not significant risk factors for opisthorchiasis (Table 4).

## Discussion

To understand the current endemicity of *O. viverrini*, this study applied the urine antigen assay, in combination with fecal egg examination, to a large-scale screening of *O. viverrini* in 21 provinces in Thailand. Our results highlight that the prevalence of *O. viverrini* by urine assay was approximately four-fold higher than those determined by FECT across all study areas (12.2% by FECT and 50.3% by urine antigen). Compared with the composite reference standard, the diagnostic sensitivity of urine antigen assay (91.6%) was about four-fold higher than that for FECT (21.9%). The high geographical variation in prevalence by province (range 20–71%), which includes areas of high endemicity (prevalence >60%), observed by the urine assay in some localities is of particular concern and poses a challenge for future management and control of parasite transmission.

In this study, a substantial proportion of participants (49.2%) who tested negative for *Opisthorchis viverrini* infection by FECT were positive by urine ELISA. This discrepancy may reflect potential false-positive results, possibly due to cross-reactivity with antigens from other parasitic infections such as *Strongyloides stercoralis*, hookworms, and minute intestinal flukes (MIFs). Additionally, the higher sensitivity of the ELISA method compared to FECT may also contribute to the observed discordance and this finding is consistent with our previous reports in northeast Thailand [19–21]. Conversely, some individuals were fecal egg-positive but urine antigen-negative, which also aligned with findings from earlier reports [19,20]. This discrepancy may result from the misidentification of minute intestinal flukes as *O. viverrini* or other underlying factors which restrict the passage of eggs into stool, such as biliary pathology, require further investigation.

We previously showed that urine assay is more sensitive than fecal examination, and a positive diagnosis by urine antigen detection indicates active or current *O. viverrini* infection in humans and also in experimental laboratory animal studies [22,27]. There are several lines of evidence supporting the reliability of the urine antigen assay. First, the antigen levels in opisthorchiasis-infected individuals diminished after curative treatment in either fecal egg-positive or negative individuals [27]. After curative treatment, the antigen in urine reemerged because of reinfection by *O. viverrini* in some individuals 10 months later [28]. Second, the findings of comparable profiles of age-sex-prevalence and intensity of *O. viverrini* infection between FECT and urine assay demonstrated the utility and reliability of urine assay in the diagnosis

PLOS Neglected Tropical Diseases

**Table 4. Demographic and lifestyle as risk factors for *O. viverrini* infection determined by FECT and urine ELISA in 21 provinces of Thailand using univariate and multivariate logistic regression (n=16,701).**

| Factors | No. examined | *O. viverrini* positive n (%) | negative n (%) | OR | aOR | 95%CI | p-value |
|---|---|---|---|---|---|---|---|
| **Gender** | | | | | | | |
| Females | 9,767 | 5,016 (51.4) | 4,751 (48.6) | 1 | 1 | | |
| Males | 6,934 | 4,232 (61.0) | 2,702 (39.0) | 1.48*** | 1.49 | 1.39–1.59 | **<0.001** |
| **Age groups (years)** | | | | | | | |
| < 20 | 477 | 246 (51.6) | 231 (48.4) | 1 | 1 | | |
| 20–29 | 522 | 249 (47.7) | 273 (52.3) | 0.86 | 0.87 | 0.67–1.11 | 0.254 |
| 30–39 | 1,525 | 795 (52.1) | 730 (47.9) | 1.02 | 1.05 | 0.85–1.29 | 0.637 |
| 40–49 | 4,402 | 2,532 (57.5) | 1,870 (42.5) | 1.27* | 1.31 | 1.08–1.58 | **0.006** |
| 50–59 | 5,081 | 2,870 (56.5) | 2,211 (43.5) | 1.24* | 1.26 | 1.05–1.53 | **0.015** |
| 60–69 | 3,554 | 1,915 (53.9) | 1,639 (46.1) | 1.10 | 1.11 | 0.91–1.34 | 0.306 |
| ≥70 | 1,002 | 542 (54.1) | 460 (45.9) | 1.11 | 1.11 | 0.89–1.38 | 0.353 |
| **Educational levels** | | | | | | | |
| Primary and lower | 12,362 | 6,904 (55.8) | 5,458 (44.2) | 1 | 1 | | |
| Secondary | 3,619 | 1,991 (55.0) | 1,628 (45.0) | 0.95 | 0.94 | 0.86–1.03 | 0.199 |
| Certificate and higher | 720 | 353 (49.0) | 367 (51.0) | 0.76*** | 0.80 | 0.67–0.95 | **0.013** |
| **Occupational status** | | | | | | | |
| Non-farmer | 3,389 | 1,781 (52.6) | 1,608 (47.4) | 1 | 1 | | |
| Farmer | 13,312 | 7,467 (56.1) | 5,845 (43.9) | 1.15*** | 1.14 | 1.04–1.24 | **0.004** |
| **History of *O. viverrini* examination** | | | | | | | |
| Yes, and undetected | 3,792 | 1,976 (52.1) | 1,816 (47.9) | 1 | 1 | | |
| Never | 11,138 | 6,251 (56.1) | 4,887 (43.9) | 1.18*** | 1.12 | 1.06–1.19 | **<0.001** |
| Yes, and found the *O. viverrini* eggs | 1,771 | 1,021 (57.7) | 750 (42.3) | 1.25*** | 1.24 | 1.14–1.34 | **<0.001** |
| **History of praziquantel treatment** | | | | | | | |
| Never | 14,035 | 7,691 (54.8) | 6,344 (45.2) | 1 | 1 | | |
| Once and over | 2,666 | 1,557 (56.4) | 1,109 (41.6) | 1.16** | 1.21 | 1.06–1.39 | **0.006** |
| **Family history with liver cancer** | | | | | | | |
| No | 15,417 | 8,570 (55.6) | 6,847 (44.4) | 1 | 1 | | |
| Yes | 1,284 | 678 (52.8) | 606 (47.2) | 0.90 | 1.02 | 0.89–11.64 | 0.797 |
| **Smoking history** | | | | | | | |
| No | 13,446 | 7,261 (54.0) | 6,185 (46.0) | 1 | 1 | | |
| Yes, current, or previous | 3,255 | 1,987 (61.0) | 1,268 (39.0) | 1.34*** | 1.05 | 0.95–1.16 | 0.342 |
| **Alcohol consumption history** | | | | | | | |
| No | 10,120 | 5,475 (54.1) | 4,645 (45.9) | 1 | 1 | | |
| Yes, current, or previous | 6,581 | 3,773 (57.3) | 2,808 (42.7) | 1.14*** | 1.01 | 0.94–1.08 | 0.857 |
| **History of raw fish eating** | | | | | | | |
| No | 3,349 | 1,784 (53.3) | 1,565 (46.7) | 1 | 1 | | |
| Yes, current, or previous | 13,352 | 7,464 (55.9) | 5,888 (44.1) | 1.11** | 1.11 | 1.02–1.20 | **0.011** |

Data presented were analyzed by logistic regression model showing OR and aOR with 95% CI and p-values.

*, **, *** indicate OR with a significance level of $P<0.05$, $P<0.01$ and $P<0.001$, respectively.

of opisthorchiasis [20]. Third, evidence from animal models indicated that the antigen detected in urine is directly linked to the presence of adult worms in the liver [22]. Finally, unlike fecal examination, the urine assay had low inter-day variation in positive detection rates (0.8%-2.2%) compared with fecal examination, with a greater variation from 7.5%-10.0% [21]. The inter-day variability of fecal examination might be due to the inconsistency of egg excretion by the adult worms, particularly in the case of low worm burden. These aforementioned data indicate that a single urine test is reliable for the diagnosis of opisthorchiasis when compared with a fecal examination on either one or repeated over three days.

Morbidity arising from infection with *O. viverrini*, such as hepatobiliary disease and cholangiocarcinoma, depends on the intensity of infection, which is caused by the underlying worm burden within hosts [6,29]. The intensity of infection among *O. viverrini*-positive individuals for fecal egg counts (mean EPG = 16.9) and urine antigen concentration (mean = 164 ng/mL urine) observed in this study were apparently low due to the aggregated distribution pattern within the participants, which is consistent with a negative binomial distribution of adult *O. viverrini* worms within hosts [8]. It is anticipated that through long-standing control by mass drug administration (MDA) and socioeconomic development [30], the intensity of infection of *O. viverrini* should be gradually reduced compared with the previous reports in 1980 [31,32]. Under the scenario of widespread light infection, the reliability of a single fecal examination needs cautious interpretation [19,21]. Indeed, the outcome of the Mass Drug Administration (MDA) and transmission dynamics of *O. viverrini* were accurately demonstrated in some well-defined cohort studies using the conventional fecal examination method [31–33]. Thus, the availability of urine assay, which is sensitive to detect light infection, can contribute to an accurate assessment of the current transmission, and this strengthens the effectiveness of the control programs. Further investigation is required to determine whether a single-worm infection can be detected by urine assay since this is a critical threshold to assess the impact of parasite control and elimination programs.

Currently, the pathway in which *O. viverrini* antigen originates from the bile ducts and eventually appears in urine is not clearly understood. Previous studies have noted that there was a pathological change and the presence of immune complexes in the kidneys of *O. viverrini*-infected hamsters [34], and *O. viverrini* antigen was detected in inflamed biliary tissue, where it became more pronounced after drug treatment in the hamster model [35]. In humans, the presence of *O. viverrini* antigen in kidney tissue has been implicated in chronic kidney disease [36]. Although further work on the pathway and circulation of *O. viverrini* antigen deserved more study, we anticipated that antigen produced by worms in the bile ducts enters blood circulation and is filtered through the kidneys before being excreted in the urine.

The results of the analysis of risk factors for opisthorchiasis revealed traditional roles of demographic data such as the age and sex of the participants [37]. The profiles of age-prevalence and intensity of infection in *O. viverrini* infection suggested a low and repeated exposure to infection arose from fish consumption over time [6]. This assumption is supported by the low incidence of infection and reinfection [38,39]. We also observed low reinfection rates in a cohort population in northeast Thailand [27]. The results that males have a higher risk of infection when assessed by fecal examination and urine assay are in agreement with previous reports [40]. The result for the history of praziquantel treatment was unexpected since those who received one or more treatments indicated reinfection or individuals with a high risk of infection, such as fishermen, and this was similar to a previous report [41]. Smoking and alcoholic drinks, as well as raw fish consumption, are often dominated by males, hence a greater prevalence of infection was found in males compared to females. The analysis of risk factors for *O. viverrini* infection in this study indicated once again the multiple classical risk factors such as age, sex, occupation, history of praziquantel treatment, history of *O. viverrini* examination, and raw fish consumption. The findings that education level and non-farmer occupations are associated with the prevention of opisthorchiasis are encouraging, as many farmers in this study lacked primary education, highlighting the significant role of educational background in liver fluke transmission. This aligns with previous advocacy [42], suggesting that education plays a central role in preventing opisthorchiasis.

There are several challenges and limitations in this study. First, due to the nature of a large-scale screening study, a single urine and fecal sample were analyzed, and under-diagnosis, particularly those by fecal examination, is expected

[21]. Second, the occurrence of false-positive results in opisthorchiasis diagnosis using urine ELISA, when compared to FECT, underscores the necessity for further investigation. Future research should focus on evaluating urinary antigen detection across diverse endemic settings. Third, fecal examination is the conventional gold standard for diagnosing opisthorchiasis and other helminth infections. However, an autopsy study demonstrated that worm recovery of *Opisthorchis viverrini* in the biliary system is more sensitive than fecal egg examination, as the majority of individuals with fewer than 20 worms had no detectable eggs in their feces [43]. Therefore, recovering worms is a more accurate method for diagnosing opisthorchiasis than fecal examination. However, both fecal egg and antigen detection indicate the presence of worms in the liver, i.e., active opisthorchiasis. Fourth, since each province may have several districts and hundreds of sub-districts, the prevalence data arise from one sub-district per province, and the data may not directly infer to the provincial levels. The majority of participants in the project were from northeast Thailand, and a small percentage were from other regions, i.e., the north and central Thailand. Hence, the level of *O. viverrini* transmission in these regions of Thailand required more studies in a wider locality. Another limitation is the exclusion of participants aged < 15 years due to infection risk, leaving a gap in understanding opisthorchiasis transmission in younger age groups. Although the urine assay by ELISA had no cross-reaction with minute intestinal fluke, such as *Haplorchis* spp., confirmation by PCR was not done in our study. The presence of mixed *O. viverrini* and *Haplorchis* spp. was shown by PCR in some provinces in northern Thailand [44]. Lastly, the analysis platform by ELISA is not suitable for resource-poor settings. A rapid diagnostic test for point-of-care use is clearly more suitable [45] and was not available at the time of conducting this project.

## Conclusion

From a large-scale survey of opisthorchisis covering 21 provinces in this study, the overall prevalence by urinary antigen assay was 50.3%, which was four times higher than observed by FECT (12.2%). Several high transmission localities with a prevalence >60% were identified, and these posed a challenging task for a more effective strategy for the prevention and control of opisthorchiasis. The quantitative correlations between the intensity of infection (fecal egg counts) and antigen concentration in urine suggest the quantitative advantage of urine assay. The prevalence of *O. viverrini* by both fecal exam and urine assay was dominant in males compared with females and the prevalence profile increased with age, which suggests a repeat and low-dose infection pattern. Similar aggregate distributions of fecal egg counts and urine antigen concentration exhibited typical distribution patterns in helminths in their hosts and were consistent with the negative binomial distribution. Analysis of risk factors for *O. viverrini* infection showed that apart from age, sex, history of drug treatment, and raw fish consumption, education levels and occupations emerged as protective factors against *O. viverrini* infection. The baseline data created in this study forms the basis for a cohort population, which aims to develop a disease control program that includes repeated screening and annual treatment for opisthorchiasis.

## Supporting information

**S1 Text. Sample size calculation [46].**
(DOCX)

**S1 Fig. Comparative prevalence rates of *O. viverrini* infection by FECT and urine antigen assays in 21 provinces of Thailand (n = 18,062).** Dark bars indicate prevalence as determined by FECT; superimposed white bars indicate prevalence measured using urinary antigen detection by ELISA. Bars are arranged in order of ascending prevalence based on FECT results. CMI: Chiang Mai, NBP: Nong Bua Lamphu, LEI: Loei, UBN: Ubon Ratchathani, SKW: Sa Kaeow, PYO: Phayao, BKN: Bueng Kan, CPM: Chaiyaphum, UDN: Udon Thani, KKN: Khon Kaen, NKI: Nong Khai, SRN: Surin, RET: Roi Et, NMR: Nakhon Ratchasima, SSK: Si Sa Ket, YST: Yasothon, MKM: Maha Sarakham, BRM: Buriram, NPM: Nakhon Phanom, SNK: Sakon Nakhon, and KSN: Kalasin.
(TIF)

**S2 Fig. Frequency distribution of *O. viverrini* antigen and egg in feces.** The fecal egg counts frequency (median = 16.9 EPG) (A), and the urine antigen concentration (median = 47.8 ng/ml) among antigen-positive individuals (B) within the sample population in Thailand.
(TIF)

**S1 Checklist. STARD Checklist.**
(DOC)

**S1 Data. Files.**
(XLSX)

## Acknowledgments

For the purpose of open access, the author has applied a CC BY public copyright license to any Author Accepted Manuscript version arising from this submission.

## Author contributions

**Conceptualization:** Kulthida Y. Kopolrat, Jiraporn Sithithaworn, Watcharin Loilome, Paiboon Sithithaworn.

**Data curation:** Kulthida Y. Kopolrat, Phattharaphon Wongphutorn, Anchalee Techasen, Chatanun Eamudomkarn, Jiraporn Sithithaworn, Paiboon Sithithaworn.

**Formal analysis:** Kulthida Y. Kopolrat, Phattharaphon Wongphutorn, Thomas Crellen, Paiboon Sithithaworn.

**Funding acquisition:** Watcharin Loilome, Nisana Namwat, Attapol Titapun, Naraong Khuntikeo, Thomas Crellen, Paiboon Sithithaworn.

**Investigation:** Kulthida Y. Kopolrat, Chanika Worasith, Paiboon Sithithaworn.

**Methodology:** Kulthida Y. Kopolrat, Chanika Worasith, Nisana Namwat, Thomas Crellen, Paiboon Sithithaworn.

**Project administration:** Watcharin Loilome, Samarn Futrakul, Naraong Khuntikeo, Paiboon Sithithaworn.

**Resources:** Kulthida Y. Kopolrat, Anchalee Techasen, Chatanun Eamudomkarn, Nisana Namwat, Attapol Titapun, Bandit Thinkhamrop, Thomas Crellen.

**Software:** Kulthida Y. Kopolrat, Phattharaphon Wongphutorn, Chaiwat Tawarungruang, Thomas Crellen.

**Supervision:** Jiraporn Sithithaworn, Watcharin Loilome, Nisana Namwat, Attapol Titapun, Bandit Thinkhamrop, Naraong Khuntikeo, Thomas Crellen, Paiboon Sithithaworn.

**Validation:** Kulthida Y. Kopolrat, Chanika Worasith, Thomas Crellen, Paiboon Sithithaworn.

**Visualization:** Kulthida Y. Kopolrat, Chatanun Eamudomkarn, Paiboon Sithithaworn.

**Writing – original draft:** Kulthida Y. Kopolrat, Naraong Khuntikeo, Thomas Crellen, Paiboon Sithithaworn.

**Writing – review & editing:** Samarn Futrakul, Simon D Taylor-Robinson, Melissa R Haswell, Thomas Crellen, Paiboon Sithithaworn.

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
