## [Decision Letter · Decision Letter 0]

PNTD-D-24-01483

Large-scale epidemiology of opisthorchiasis in 21 provinces in Thailand based on diagnosis by fecal egg examination and urine antigen assay and analysis of risk factors for infection

Dear Dr. Sithithaworn,

Thank you for submitting your manuscript to PLOS Neglected Tropical Diseases. After careful consideration, we feel that it has merit but does not fully meet PLOS Neglected Tropical Diseases's publication criteria as it currently stands. Therefore, we invite you to submit a revised version of the manuscript that addresses the points raised during the review process.

Please submit your revised manuscript within 60 days Feb 17 2025 11:59PM. If you will need more time than this to complete your revisions, please reply to this message or contact the journal office at plosntds@plos.org. Please include the following items when submitting your revised manuscript:

We look forward to receiving your revised manuscript.

Kind regards,

Javier Sotillo

Academic Editor

Francesca Tamarozzi

Section Editor

Shaden Kamhawi

co-Editor-in-Chief

Paul Brindley

co-Editor-in-Chief

**Additional Editor Comments:**

In the basence of a gold standard, the ideal analysis is not comparing a test result with a composite reference but a latent class analysis. We prompt the authors to explore whether they might have the capacity to implement such analysis. Since FECT cannot be considered a gold standard, one of the reviewers recommended to perform PCR, however, this neither can be considered a gold standard. We prompt the authors to consider if possible performing also a PCR on their samples. If not possible, we prompt the authors to declare why and include these considerations in the discussion, among the limits of the study. In the latter case, we promt the authors to consider at least providing a 2x2 evaluation of FECT and antigen test results, in order to show and comment the senistivity of the antigen test, which theoretically should detect all FECT-positive samples.

However, I do not think, as suggested by the reviewer (bullet 4), that PCR on feces would be a better "reference standard" than FECT. Regarding the 3rd bullet of the last point, I am not sure if he wants to consider only FECT as the gold standard (I hope not) or if (and I would agree) is asking about the evidence that the antigen test is actually positive in all FECT-pos sample.

What do you think we write an "Editor request" (of course, if you agree with my observations) where we ask to comment on the limits of using a composite than a latent class and to interpret the 3rd bullet of the last point as a request for providing actual 2x2 table Ag vs FECT to double check this point?

**Journal Requirements:**

At this stage, the following Authors/Authors require contributions: Kulthida Y Kopolrat, Chanika Worasith, Phattharaphon Wongphutorn, Anchalee Techasen, Chatanun Eamudomkarn, Jiraporn Sithithaworn, Watchalin Loilome, Nisana Namwat, Attapol Titapun, Chaiwat Tawarungruang, Bandit Thinkhamrop, Samarn Futrakul, Simon D Taylor-Robinson, Melissa R Haswell, Thomas Crellen, and Paiboon Sithithaworn. Please ensure that the full contributions of each author are acknowledged in the "Add/Edit/Remove Authors" section of our submission form.

2) Tables should not be uploaded as individual files. Please remove these files and include the Tables in your manuscript file as editable, cell-based objects. For more information about how to format tables, see our guidelines:

https://journals.plos.org/plosntds/s/tables

3) We have noticed that you have cited Table 5 in the manuscript file but there is no corresponding table in the manuscript. Please amend your manuscript to include this table noting that tables should not be uploaded as individual files.

4) We note that your Data Availability Statement is currently as follows: "The data that support the findings of this study are available from the corresponding author upon reasonable request.". Please confirm at this time whether or not your submission contains all raw data required to replicate the results of your study. Authors must share the “minimal data set” for their submission. PLOS defines the minimal data set to consist of the data required to replicate all study findings reported in the article, as well as related metadata and methods (https://journals.plos.org/plosone/s/data-availability#loc-minimal-data-set-definition).

- The points extracted from images for analysis..

5) Please amend your detailed Financial Disclosure statement. This is published with the article. It must therefore be completed in full sentences and contain the exact wording you wish to be published. Please ensure that the funders and grant numbers match between the Financial Disclosure field and the Funding Information tab in your submission form. Note that the funders must be provided in the same order in both places as well.

**Reviewers' Comments:**

Reviewer's Responses to Questions

**Key Review Criteria Required for Acceptance?**

**Methods**

-Are the objectives of the study clearly articulated with a clear testable hypothesis stated?

-Is the study design appropriate to address the stated objectives?

-Is the population clearly described and appropriate for the hypothesis being tested?

-Is the sample size sufficient to ensure adequate power to address the hypothesis being tested?

-Were correct statistical analysis used to support conclusions?

-Are there concerns about ethical or regulatory requirements being met?

Reviewer #1: (No Response)

Reviewer #2: I recommend a minor revision to this manuscript, as I do not consider that new investigations, analysis, or experiments are needed in this study.

Regarding the Methods section:

- The objectives are clear and the study design seems appropriate to address the objectives.

- The population is well described and the sample size well justified. Laboratory methods, questionnaire and statistical analysis are also well described and structured with the Results section.

- Descriptions in the Methods section ensure repeatability/reproducibility of the study, especially the laboratory methods.

I wonder whether the following details could be explained/included:

- Why was the age group younger than 15 years excluded from the study? This could be useful, especially as the recruited cohort of this cross-sectional study is intended to serve as the basis of a longitudinal study (if I understood correctly).

- Urine samples: If possible, I suggest expressing centrifugation units in RCF (‘g’) instead of rpm, as it is more informative. Line 247 for FECT has both rpm and RCF.

- Urine samples were frozen (-20 °C) until analysis: Could freezing have an impact on antigen ELISA quantifications? Do similar studies all use frozen urine, or do some of these use freshly collected (or cooled only) urine samples?

**Results**

-Does the analysis presented match the analysis plan?

-Are the results clearly and completely presented?

-Are the figures (Tables, Images) of sufficient quality for clarity?

Reviewer #1: (No Response)

Reviewer #2: - The structure and presentation of results is overall quite clear, succint, and aligned with what is presented in the Methods section. For example, I find figures Fig 2 and 6 very illustrative and informative while keeping it simple, and I would like to thank the authors for it.

Some details to consider:

- Check consistency in reported overall prevalence based on urine antigen ELISA test (50.3% in Table 1, line 478, line 564; vs. 50.9% in lines 72, 109, 329). Based on numbers of Table 1, 50.3% seems to be correct.

- Line 327: Do the authors maybe prefer to mention the lowest (3.6%) and highest prevalence by FECT?

- Line 330: Highest prevalence by antigen ELISA test is 59.3% in the East region, based on Table 1.

- Could the authors elaborate, and possibly include in the Methods or Results section, how the composite reference standard assigned disease status based on FECT and antigen results? Should both diagnostics be positive to classify an individual as positive for opistorchiasis?

- Line 431: “sensitivity for urine ELISA was 91.1%” vs. Table 3 (91.6%).

- Line 444: It should refer to Table 4, not Table 3.

- Lines 461-462 (educational level) is somewhat redundant with lines 447-449 and could be combined to include aOR data.

- S Fig 1 and S Fig 2 appear to be the same. I believe S Fig 1 is missing.

- Fig 5: Is Log10 transformation needed, or the best choice for this figure? Would concentration in ng/mL show a more clear increasing trend as intensity of infection (EPG) increases? Or would it be too skewed?

**Conclusions**

-Are the conclusions supported by the data presented?

-Are the limitations of analysis clearly described?

-Do the authors discuss how these data can be helpful to advance our understanding of the topic under study?

-Is public health relevance addressed?

Reviewer #1: (No Response)

Reviewer #2: - Conclusions are brief, summarise the results well, and are supported by the data presented.

I would like to suggest the following in the Discussion section:

- Lines 535-538 and 545-547 could be combined to avoid redundancy.

- The authors are aware and transparent about the limitations of the study, and this is always appreciated from a reader's perspective. I would maybe add the age group below 15 years old as either a limitation of the study, or an explanation of why this age group was not recruited, if intentional (if the latter, likely in the Methods section instead).

- While brief, the authors do address the public health relevance and next steps (a longitudinal study).

**Editorial and Data Presentation Modifications?**

Reviewer #1: (No Response)

Reviewer #2: - Lines 140-144: I would recommend splitting this long sentence in two, or rephrase it to make the reading flow smoother.

- Line 167: Change “we” for “We” after full stop.

- Consistency in the use of singular vs plural verbs (was/were) after measurement units: Line 244 “Two grams of fresh stool was processed”, vs line 246, “Three milliliters of ethyl acetate were added”. This might be up to the journal’s guidelines rather than the authors.

- Lines 252-254: I believe a verb is missing at the end of this sentence, e.g., “was performed based on…”.

- Line 292: The dot is in red colour.

- Line 300: “(iii)” is written twice and “(ii)” is missing.

- Line 334: I would suggest removing the comma in “the roundworm, Strongyloides stercoralis”

- Grams abbreviation: I would recommend “g” instead of “gm”, unless the journal recommends otherwise (lines 390 and 409).

- Line 421: Change “mlL” for “mL”.

- Line 422: I believe “<100 µg/mL” should read “100 ng/mL”

- Line 425: For consistency, I recommend changing “epg” for “EPG”.

- Line 469: “This is the Table 5 legend” should be removed.

- Line 478: Add % symbol to 91.6.

- Line 479: I suggest change “fourfold” for “four-fold”.

- Line 519: Should “originated” read “originates”?

**Summary and General Comments**

Reviewer #1: (No Response)

Reviewer #2: This article describes a cross-sectional study to better understand the epidemiology of opistorchiasis in 21 provinces in Thailand, and serving as the basis of a future longitudinal study that will help in the national programme for the control and elimination of this NTD.

By using diagnostic methods (egg microscopy and a labortory-based ELISA test to detect adult worm antigen in urine, the latter being a superior diagnostic) as well as a questionnaire, the prevalence and the risk factors associated with infection were analysed, respectively.

The Introduction is clear and short enough to provide the background information needed, while swiftly leading the reader into the topic.

The addition of a STROBE checklist for cross-sectional studies assures the authors have thoroughly made sure all or most of the relevant information is included.

One maybe minor comment, is that there seems to be no specific funding for this work (Financial disclosure section at the beginning of the submitted manuscript), while there are authors listed in the Authors Contribution section, with Funding acquisition roles, and funding from Wellcome Trust (Grant number 215919/Z/19/Z) in the Acknowledgements section.

I would like to thank the authors for a well written, well structured, and to-the-point manuscript, on a topic of relevance and with direct application for control and elimination programmes. One of the limitations the authors mention is the lack of field-deployability of the antigen ELISA test used in this study, and I wish them all the success in trying to widely deploy the rapid antigen test for the point-of-care detection of opistorchiasis infection mentioned in Worasith et al. (2023).

PLOS authors have the option to publish the peer review history of their article (what does this mean? ). If published, this will include your full peer review and any attached files.

**Do you want your identity to be public for this peer review?** For information about this choice, including consent withdrawal, please see our Privacy Policy .

Reviewer #1: No

Reviewer #2: **Yes: ** Elías Kabbas-Piñango

**Figure resubmission:**

**Reproducibility:**



---

## [Decision Letter · Decision Letter 1]

Response to Reviewers
Revised Manuscript with Track Changes
Manuscript

Shaden Kamhawi

co-Editor-in-Chief

Paul Brindley

co-Editor-in-Chief

**Reviewers' comments:**

**Key Review Criteria Required for Acceptance?**

**Methods:**

-Are the objectives of the study clearly articulated with a clear testable hypothesis stated?

-Is the study design appropriate to address the stated objectives?

-Is the population clearly described and appropriate for the hypothesis being tested?

-Is the sample size sufficient to ensure adequate power to address the hypothesis being tested?

-Were correct statistical analysis used to support conclusions?

-Are there concerns about ethical or regulatory requirements being met?

Reviewer #1: A revision was made as suggested. The way the authors use standalone FECT and composite results is acceptable.

Reviewer #2: No further comments regarding the Methods section.

**Results:**

-Does the analysis presented match the analysis plan?

-Are the results clearly and completely presented?

-Are the figures (Tables, Images) of sufficient quality for clarity?

Reviewer #1: Results are now better stratified (age intervals corrected to 10-year gaps). Analysis more clearly distinguishes the groups (FECT+/Urine+, FECT-/Urine-, etc.), but critical evaluation is limited when discordant results occur (FECT-/Urine+). The authors might need to state this limitation in the manuscript.

Reviewer #2: - Line 428 still shows 50.9% in positive rate for urine ELISA. Should this be 50.3%, same as the reported prevalence by urine ELISA?

- Table 3: Please revise numbers in the 2x2, as not all of them add up to the total referred. The previous table had some diagnostic accuracy parameters (sensitivity, specificity, PPV, NPV...). Could the authors include (some of) them, as they were in the previous version, for each scenario, “a” and “b”?

**Conclusions:**

-Are the conclusions supported by the data presented?

-Are the limitations of analysis clearly described?

-Do the authors discuss how these data can be helpful to advance our understanding of the topic under study?

-Is public health relevance addressed?

Reviewer #1: Conclusions slightly softened, but the potential for overstatement of urine assay performance persists without robust justification. The authors might still need to state this limitation in the manuscript.

Reviewer #2: No further comments regarding the Conclusions section.

**Editorial and Data Presentation Modifications?**

Reviewer #1: (No Response)

Reviewer #2: (No Response)

**Summary and General Comments:**

Reviewer #1: The work remains highly novel (the first large-scale urine antigen epidemiological survey across Thailand) and impactful.

Reviewer #2: (No Response)

PLOS authors have the option to publish the peer review history of their article (what does this mean? ). If published, this will include your full peer review and any attached files.

**Do you want your identity to be public for this peer review?** For information about this choice, including consent withdrawal, please see our Privacy Policy .

Reviewer #1: No

Reviewer #2: **Yes: ** Elías Kabbas-Piñango

**Figure resubmission:**

**Reproducibility:** To enhance the reproducibility of your results, we recommend that authors of applicable studies deposit laboratory protocols in protocols.io, where a protocol can be assigned its own identifier (DOI) such that it can be cited independently in the future. Additionally, PLOS ONE offers an option to publish peer-reviewed clinical study protocols. Read more information on sharing protocols at https://plos.org/protocols?utm_medium=editorial-email&utm_source=authorletters&utm_campaign=protocols

---

## [Editor Report · Decision Letter 2]

Dear Professor Sithithaworn,

We are pleased to inform you that your manuscript 'Large-scale epidemiology of opisthorchiasis in 21 provinces in Thailand based on diagnosis by fecal egg examination and urine antigen assay and analysis of risk factors for infection' has been provisionally accepted for publication in PLOS Neglected Tropical Diseases.

Best regards,

Javier Sotillo

Academic Editor

Francesca Tamarozzi

Section Editor

Shaden Kamhawi

co-Editor-in-Chief

Paul Brindley

co-Editor-in-Chief

---

## [Editor Report · Acceptance letter]

Dear Professor Sithithaworn,

We are delighted to inform you that your manuscript, "Large-scale epidemiology of opisthorchiasis in 21 provinces in Thailand based on diagnosis by fecal egg examination and urine antigen assay and analysis of risk factors for infection," has been formally accepted for publication in PLOS Neglected Tropical Diseases.

Best regards,

Shaden Kamhawi

co-Editor-in-Chief

Paul Brindley

co-Editor-in-Chief
